# Prediction of HIV status based on socio-behavioural characteristics in East and Southern Africa

Erol Orel[1]*, Rachel Esra[1], Janne Estill[1,2], Amaury Thiabaud[1], Stéphane Marchand-Maillet[3], Aziza Merzouki[1], Olivia Keiser[1]

**1** Institute of Global Health, University of Geneva, Geneva, Switzerland, **2** Institute of Mathematical Statistics and Actuarial Science, University of Bern, Bern, Switzerland, **3** Department of Computer Science, Viper Group, University of Geneva, Geneva, Switzerland

☯ These authors contributed equally to this work.
* Erol.Orel@unige.ch

## Abstract

### Introduction

High yield HIV testing strategies are critical to reach epidemic control in high prevalence and low-resource settings such as East and Southern Africa. In this study, we aimed to predict the HIV status of individuals living in Angola, Burundi, Ethiopia, Lesotho, Malawi, Mozambique, Namibia, Rwanda, Zambia and Zimbabwe with the highest precision and sensitivity for different policy targets and constraints based on a minimal set of socio-behavioural characteristics.

### Methods

We analysed the most recent Demographic and Health Survey from these 10 countries to predict individual's HIV status using four different algorithms (a penalized logistic regression, a generalized additive model, a support vector machine, and a gradient boosting trees). The algorithms were trained and validated on 80% of the data, and tested on the remaining 20%. We compared the predictions based on the F1 score, the harmonic mean of sensitivity and positive predictive value (PPV), and we assessed the generalization of our models by testing them against an independent left-out country. The best performing algorithm was trained on a minimal subset of variables which were identified as the most predictive, and used to 1) identify 95% of people living with HIV (PLHIV) while maximising precision and 2) identify groups of individuals by adjusting the probability threshold of being HIV positive (90% in our scenario) for achieving specific testing strategies.

### Results

Overall 55,151 males and 69,626 females were included in the analysis. The gradient boosting trees algorithm performed best in predicting HIV status with a mean F1 score of 76.8% [95% confidence interval (CI) 76.0%-77.6%] for males (vs [CI 67.8%-70.6%] for SVM) and 78.8% [CI 78.2%-79.4%] for females (vs [CI 73.4%-75.8%] for SVM). Among the ten most

**Data Availability Statement:** The data are to be found on the DHS website: the data are held in a public repository: https://dhsprogram.com/data/available-datasets.cfm.

**Funding:** We acknowledge the support of the Swiss National Science Foundation (SNF professorship grant n˚ 163878 and grant n˚ 202660 to O Keiser) which funded this study. The funders had no role in study design, data collection and analysis, decision to publish, or preparation of the manuscript.

**Competing interests:** No.

predictive variables for each sex, nine were identical: longitude, latitude and, altitude of place of residence, current age, age of most recent partner, total lifetime number of sexual partners, years lived in current place of residence, condom use during last intercourse and, wealth index. Only age at first sex for male (ranked 10th) and Rohrer's index for female (ranked 6th) were not similar for both sexes. Our large-scale scenario, which consisted in identifying 95% of all PLHIV, would have required testing 49.4% of males and 48.1% of females while achieving a precision of 15.4% for males and 22.7% for females. For the second scenario, only 4.6% of males and 6.0% of females would have had to be tested to find 55.7% of all males and 50.5% of all females living with HIV.

## Conclusions

We trained a gradient boosting trees algorithm to find 95% of PLHIV with a precision twice higher than with general population testing by using only a limited number of socio-behavioural characteristics. We also successfully identified people at high risk of infection who may be offered pre-exposure prophylaxis or voluntary medical male circumcision. These findings can inform the implementation of new high-yield HIV tests and help develop very precise strategies based on low-resource settings constraints.

## Introduction

In order to reach epidemic control by 2030, the Joint United Nations Programme (UNAIDS) has set fast track targets to rapidly scale up effective HIV services [1]. One of the aims is to ensure that 95% of the approximately 38 million people globally living with HIV (PLHIV) are aware of their HIV status and that 95% of those with HIV positive diagnoses are on treatment [2].

People in East and Southern Africa are disproportionately burdened by HIV, constituting more than half of the global PLHIV with 20.7 million people currently estimated to be HIV positive [2]. As of 2020, 87% of PLHIV in this region were aware of their HIV status, of whom 83% were accessing treatment [3]. In addition, 25% of new HIV infections in East and Southern Africa were concentrated among key populations such as female sex workers, men having sex with men, prisoners, and people who inject drugs [3].

HIV is transmitted within a complex network that is influenced by biological, behavioural, and social factors. In East and Southern Africa, there is large geographical variation in the distribution of the HIV epidemic [4]. In order to identify populations at a high risk of infection, global HIV prevention efforts have shifted toward optimizing resource allocation by considering geographical data as a way of increasing program impact and efficiency [5].

Modern predictive algorithms have the power to substantially enhance HIV prevention and detection, increasing the prediction capability by processing large amounts of data of a different nature. This methodology has been implemented to establish patterns of HIV risk behaviour, to optimise HIV treatment modalities, and to identify high-risk individuals for targeted interventions from a number of novel data sources [6–15].

As more PLHIV are diagnosed, finding persons with undiagnosed HIV becomes progressively more difficult and expensive. Hence, resource constraints and potential funding shortages have resulted in demands for differentiated high yield testing strategies in parallel to provider-initiated HIV testing and counselling (PITC) [14, 16, 17]. In this paper, we aim to

identify new key populations based on socio-behavioural characteristics by comparing four different prediction algorithms. These insights intend to both inform targeted case-finding strategies as well as identify high risk HIV negative individuals eligible for prevention services such as voluntary medical male circumcision (VMMC) and/or pre-exposure prophylaxis (PrEP).

## Methods

### Data

Since 1984, the Demographic and Health Surveys (DHS) program has provided technical assistance for over 400 surveys in more than 90 countries, advancing global understanding of health and population trends in developing countries [18]. DHS are nationally-representative household surveys that provide data for a wide range of monitoring and impact evaluation indicators on health and nutrition. Standard DHS surveys have large sample sizes (usually between 5,000 and 30,000 households) and are typically conducted every five years [19]. We used the most recent DHS surveys at or after 2013 of ten East and Southern African countries (S1 Table) with a generalised HIV epidemic: Angola, Burundi, Ethiopia, Lesotho, Malawi, Mozambique, Namibia, Rwanda, Zambia and, Zimbabwe.

We combined separately male and female datasets of each country with their corresponding household's geographic position and their HIV test results. We then merged the ten countries and obtained two datasets containing 68,979 males and 83,910 females with 527 and 3,213 variables respectively, since different socio-behavioural characteristics are recorded for each sex. The target variable was the HIV status of the individuals (0 for HIV negative and 1 for HIV positive). During the data pre-processing step, only individuals with positive or negative HIV status were included in the analysis; those with unknown status were discarded. We cleaned, concatenated, filtered, transformed, and aggregated the data (S2 Table). We imputed missing values, that we assumed missing at random, using multiple imputation by chained equations (MICE) (as detailed in S3 and S4 Tables) and the data were further harmonized and scaled [20, 21]. Thus, the final dataset included 55,151 males and 69,626 females with 84 and 122 variables respectively; 73 variables were common to both sexes (S5 Table).

### Training, validation and test procedure steps

**Fig 1—Step 1.** From these two datasets, we first left one of the 10 countries out (switching left out country) to create 10 different datasets per sex, each one comprised of only 9 countries. This has been done for generalization purposes in order to further assess the quality of our models when the data were not drawn from the exact same distribution. Then, each of the 10 newly created datasets per sex were split at the individual level between a stratified (due to imbalanced outcomes) 80% training set and a 20% test set. The above described MICE imputation and data standardization was then performed separately on training and test datasets to avoid information from the training dataset to contaminate the test dataset.

**Fig 1—Step 2.** Using the training datasets and 50 randomly selected sets of hyperparameters, a stratified 5-fold cross-validation was then performed for each algorithm on each of the training sets for training and validation. The set of hyperparameters that obtained the maximum mean F1 score over the validation datasets was selected.

**Fig 1—Step 3.** Each one of the 10 best models per sex and per algorithm was then ran on the corresponding test set and the resulting metric scores were averaged. We selected the algorithm with the maximum mean F1 score over the 10 test datasets. Finally, we applied each selected model on the corresponding left out country dataset.

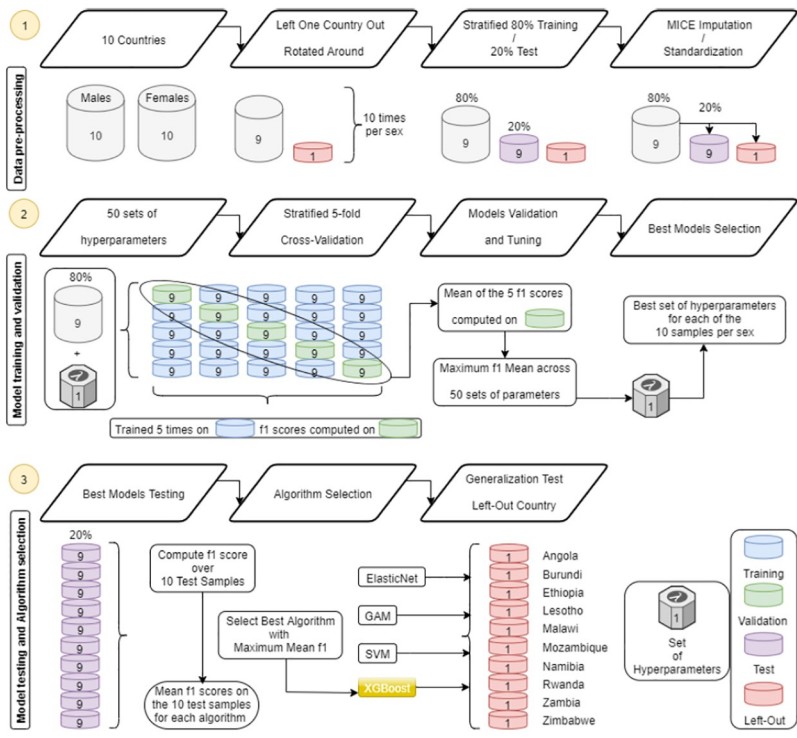

**Fig 1. Methodology diagram of the analysis part 1.**

## Algorithms

We compared four algorithms for the prediction of the HIV status of an individual: a penalized logistic regression (Elastic Net) [22], a generalized additive model (GAM) [23], a support vector machine (SVM) [24], and an implementation of gradient boosted trees (XGBoost) [25]. The Elastic Net and the GAM are among the most widely used classification methods in biology and medicine, SVM is a very common machine learning algorithm, and XGBoost is a decision-tree-based ensemble which has gained a lot of attraction since its development a few years ago due to its excellent performances (more details about the models can be found in the S1 File). Our primary interest was to find the largest number of HIV positive individuals (sensitivity) with the highest possible yield (positive predictive value (PPV)). We, therefore, used the F1 score for assessing the performance of the different algorithms. This metric combines the sensitivity and the precision in a harmonic mean and is often recommended for unbalanced datasets when comparing models [26]. The probability threshold to classify if someone is considered HIV positive was set at 50%. In addition, to validate our results with a strictly proper scoring rule, we also computed the Brier score. This score is strictly equivalent to the mean squared error as applied to predicted probabilities for unidimensional predictions.

The analysis was done in two steps for each of the four algorithms (Fig 1—Step 2, 3), and separately for males and females. Training and validation were performed using the stratified 5-fold cross-validation on the training sample with 50 different sets of hyperparameters randomly chosen from a grid (as detailed in S1 File). Among these sets, we selected the one with the highest mean F1 score, and tested the obtained model on the test sample and on the left-out country, which were not used during training and validation (Fig 1—Step 3). We selected the best algorithm based on its averaged F1 scores on the ten test samples.

## Variables selection and HIV status prediction

For variables selection and HIV status prediction, we used the exact same training, validation and testing strategy than in the first part of our analysis except that no country was left out. We split each unique dataset par sex into a stratified 80% training and validation set and a 20% test set. The best algorithm was trained and validated using a random grid-search over 250 sets of hyperparameters and a stratified 5-fold cross-validation. The first predictions were performed using all available variables. Based on the F1 scores, sensitivity, and PPV, we compared two imputation methods, namely MICE (models M1 and F1 for males and females, respectively) and a built-in method from the selected algorithm [25] (models M2 and F2).

We used a sequential forward floating selection (SFFS), which eliminates (or adds) variables based on a defined classifier performance metric, on the 80% training samples and calculated the F1 scores for different subsets of variables. We selected the subset of variables for which the F1 scores plateaued and we then assessed the direction of the association between these variables and the probability of being HIV-positive using Shapley values [27].

We retrained the best algorithm with the above defined subsets of variables (models M3 and F3) and also on a minimal subset common to both sexes (models M4 and F4). The F1 scores, the sensitivity, and the PPV were compared to the ones obtained for M1, M2, F1, and F2. With our last models based on a minimal subset common to both sexes (models M4 and F4), we further analysed the results at country level, comparing the F1, sensitivity and PPV between countries and the differences between observed and predicted prevalences.

## Scenarios

We tested two scenarios: for the first scenario, the sensitivity was set to 95%, equivalent to 95% of PLHIV knowing their status, and we reported the corresponding precision and number of individuals to be tested. For the second scenario, we identified a population for which the probability of being HIV positive was higher than 90%. We considered that these groups of individuals are targets for specific testing strategies or ideal candidates for prevention services.

## Ethical review

No ethical approval was needed for this study.

## Data and code availability

The data supporting the findings of this study are available from the DHS Program https://dhsprogram.com/. The DHS Program is authorized to distribute, at no cost, unrestricted survey data files for legitimate academic research. Registration is required to access the data.

The data was collected between 2013 and 2017 depending on each country.

All analyses were performed in Python version 3.7.4. The code is available on https://gitlab.com/Triphon/predicting_hiv_status.

## Results

Overall, 55,151 males and 69,626 females were analysed with a prevalence ranging from 0.8% among males in Ethiopia to 33.3% among females in Lesotho with a global HIV prevalence of 8.0% (4,417 individuals) for males and 11.5% (8,011 individuals) for females. Individuals aged 25 to 34 years were the largest age group, representing 35.9% of females and 31.9% of males. About two-thirds of people lived in rural areas (S6 Table).

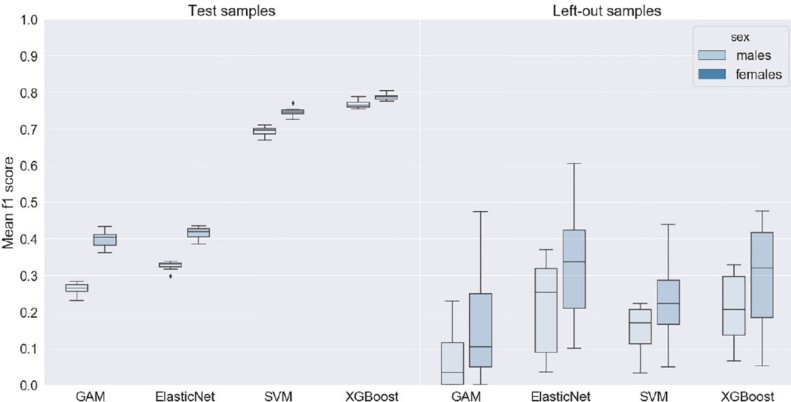

**Fig 2. Boxplot of the f1 scores for the 4 algorithms on the test and left-out samples per sex.**

### Algorithms

Fig 2 shows the performance of the four algorithms on the test samples and independent left-out countries. XGBoost had the highest F1 scores on all test samples with a mean F1 score of 76.8% [95% confidence interval (CI) 76.0%-77.6%] for males and 78.8% [CI 78.2%-79.4%] for females. In comparison, SVM had a mean F1 score of 69.2% [CI 68.2%-70.2%] for males and 74.6% [CI 73.7%-75.5%] for females. For Elastic Net, the mean F1 score was 32.6% [CI 31.8%-33.4%] for males and 41.5% [CI 40.3%-42.7%] for females. GAM performed the worst with a mean F1 score of 26.2% [CI 25.0%-27.4%] for males and 39.8% [CI 38.1%-41.5%] for females. When focusing on the Brier scores, XGBoost was still the best performing algorithm, followed by SVM, GAM and finally ElasticNet. In general, the scores obtained by the models with the best Brier score were very similar to the ones obtained with the best F1 score (S7 to S10 Tables).

When tested against the ten left-out countries, the performance of the algorithms was substantially lower than on the test samples and the F1 scores varied more widely (Fig 2—RH). The mean F1 score was the best for Elastic Net with 21.4% [CI 12.3%-30.5%] for males and 32.6% [CI 21.2%-44.0%] for females, followed closely by XGBoost with 20.9% [CI 14.3%-27.5%] and 29.8% [CI 19.0%-40.6%], respectively. In comparison, the mean F1 score for SVM was 15.4% [CI 10.9%-19.9%] for males and 22.3% [CI 14.1%-30.5%] for females. Again, GAM performed the worst with a mean F1 score of 6.6% [CI 0.9%-12.1%] and 17.1% [CI 4.4%-29.8%]. See S7 to S10 Tables for details on PPV, sensitivity and Brier scores. However, the algorithms performed generally better in countries with higher prevalence (S7 to S10 Tables).

Given that the best performance on the test samples was obtained with XGBoost, both for F1 and Brier scores, we used this algorithm for the selection of variables and the prediction of the HIV status of the individuals on the entire datasets, where no country was left out. The results on all variables using the two different imputation methods are shown in Table 1. For both sexes, the XGBoost imputation (M2 and F2) resulted in slightly higher F1 scores compared to the MICE imputation (M1 and F1). The F1 scores on the validation samples were 75.5% [CI 73.7%-77.3%] vs 74.9% [CI 73.3%-76.5%] for males and 76.1% [CI 74.9%-77.3%] vs 75.5% [CI 74.6%-76.4%] for females. Given the similarity of the obtained results, we decided to use the built-in XGBoost method for further analyses (i.e. models M3, F3, M4 and, F4) because of its simplicity of implementation and its lower computation time.

**Table 1. Results per sex of the XGBoost algorithm for different imputation methods and sets of variables.**

| | | TP | FN | FP | TN | F1 score | Sensitivity | PPV |
|---|---|---|---|---|---|---|---|---|
| Complete with MICE imputation (Model M1) | Validation | | | | | 74·9% (± 1·6%) | 71·2% (± 2·9%) | 79·1% (± 0·8%) |
| | Test | 627 | 256 | 164 | 9,984 | 74.9% | 71.0% | 79.3% |
| Complete with MICE imputation (Model F1) | Validation | | | | | 75·5% (± 0·9%) | 75·4% (± 1·6%) | 75·6% (± 0·5%) |
| | Test | 1,264 | 338 | 375 | 11,949 | 78.0% | 78.9% | 77.1% |
| Complete with XGBoost imputation (Model M2) | Validation | | | | | 75·5% (± 1·8%) | 69·6% (± 2·2%) | 82·5% (± 2·2%) |
| | Test | 617 | 266 | 122 | 10,026 | 76.1% | 69.9% | 83.5% |
| Complete with XGBoost imputation (Model F2) | Validation | | | | | 76·1% (± 1·2%) | 75·5% (± 1·7%) | 76·8% (± 1·2%) |
| | Test | 1,279 | 323 | 379 | 11,945 | 78.5% | 79.8% | 77.1% |
| 15 variables with XGBoost imputation (Model M3) | Validation | | | | | 73·7% (± 2·9%) | 67·9% (± 2·5%) | 80·7% (± 3·7%) |
| | Test | 605 | 278 | 129 | 10,019 | 74.8% | 68.5% | 82.4% |
| 27 variables with XGBoost imputation (Model F3) | Validation | | | | | 75·6% (± 1·2%) | 70·0% (± 1·2%) | 82·2% (± 1·7%) |
| | Test | 1,212 | 390 | 234 | 12,090 | 79.5% | 75.7% | 83.8% |
| 9 variables with XGBoost imputation (Model M4) | Validation | | | | | 72·9% (± 2·3%) | 65·6% (± 1·6%) | 81·9% (± 3·9%) |
| | Test | 595 | 288 | 124 | 10,024 | 74.3% | 67.4% | 82.8% |
| 9 variables with XGBoost imputation (Model F4) | Validation | | | | | 72·4% (± 1·2%) | 68·5% (± 1·4%) | 76·8% (± 1·6%) |
| | Test | 1,184 | 418 | 249 | 12,075 | 78.0% | 73.9% | 82.6% |

True Positive (TP), False Negative (FN), False Positive (FP), True Negative (TN), Positive Predictive Value (PPV)

Multiple Imputation by Chained Equations (MICE).

(± %): 95% Confidence Interval.

## Variables selection and HIV status prediction

Fig 3 shows the subset of most relevant variables to predict an individual's HIV status, as selected by the SFFS procedure. With 15 variables for males and 27 variables for females, the F1 score plateaued at 99.6% and 97%, respectively.

Among those variables, four were specific to females ('currently breastfeeding', 'fertility preference', 'time to get to water source' and 'entries in birth history') and two to males ('number of women fathered children with' and 'respondent circumcised'). Out of the ten most predictive variables for both sexes, nine were identical: geographic position (longitude, latitude, and altitude), current age, age of most recent partner, total lifetime number of sexual partners, years lived in current place of residence, condom used during last sexual intercourse with most recent partner, and a wealth index from the DHS which combines numerous wealth-related variables such as household assets and utility services [28]. The age at first sexual intercourse ranked tenth for males but only twentieth for females; the Rohrer's index (an estimate of obesity) ranked sixth for females but was not available for males.

Older age, older age of most recent partner, older age and more years since first cohabitation, higher total lifetime number of sexual partners, longer time since last sex, higher number of unions, higher number of women fathered children with, condom use during last sexual intercourse with most recent partner, having been tested for HIV, living in an urban area, higher longitude coordinate and buying vegetables from vendor with HIV were positively associated with the probability of HIV positivity for most individuals, either males, females or both. Higher age at first sex, higher wealth index, higher latitude coordinate, higher altitude, higher number of years of education, higher number of entries in birth history, circumcision, higher Rohrer's index, more years lived in place of residence, use of contraceptive, currently breastfeeding and higher number of household's members were mainly negatively associated with HIV positivity. The direction of association was not clear for the age of the household head and the time to get to the water source (Fig 3).

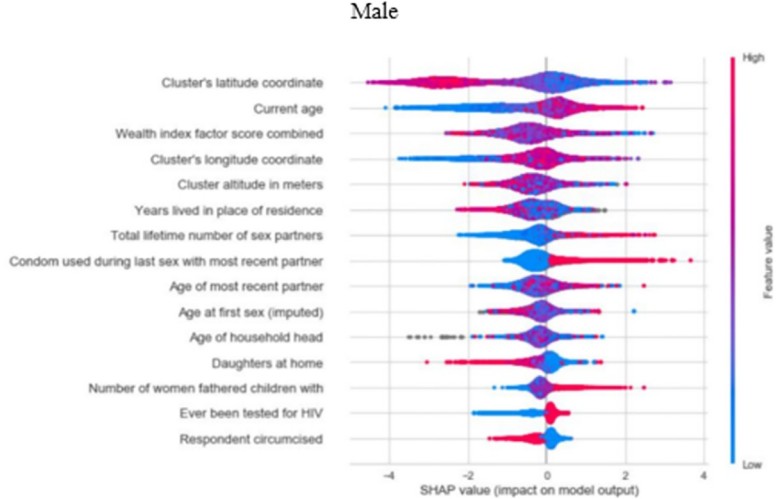

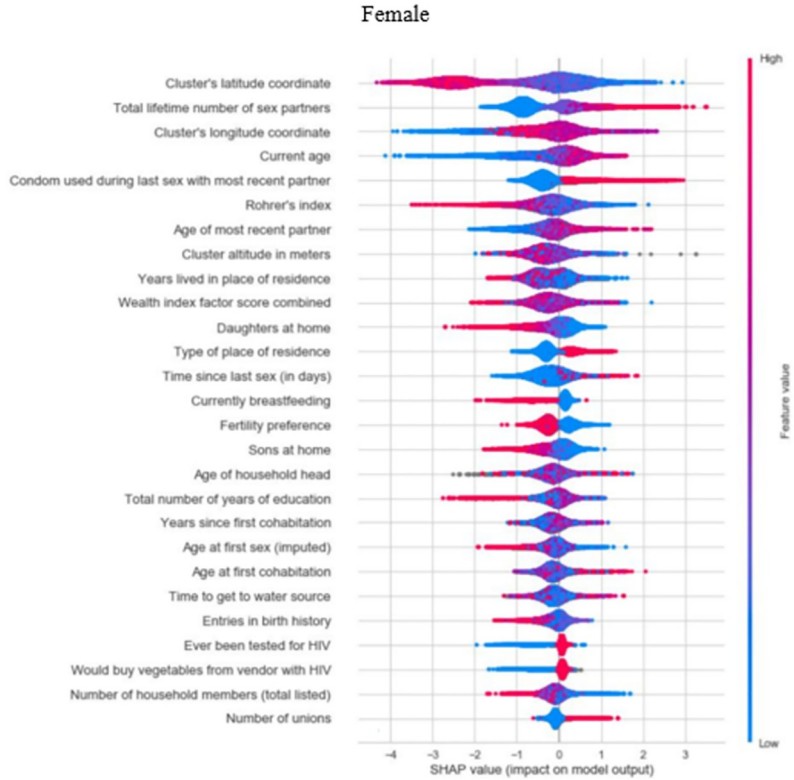

**Fig 3. Shapley values.** The variables are displayed sorted by importance from top to bottom (from the highest Shapley value to the lowest). The blue and red colours represent the value range of the variable (blue (red) represents the low (high) value range of the variable). For example, the older the age, the more likely the person will be HIV positive. N.b.: Shapley values do not describe the causal impact of each covariate, only the additional change in overall outcome by adding this covariate.

Table 1 shows the results (confusion matrix, PPV, sensitivity and F1 score) of the XGBoost algorithm on the 15 most important variables for males (M3) and 27 most important variables for females (F3). As expected from the SFFS procedure, the F1 scores of these two models were close to the scores obtained with all available variables (M2 and F2). The F1 scores decreased only by 1.8 percentage points for males and by 0.5 percentage points for females. In comparison, by using the nine most predictive common variables (M4 and F4), the F1 scores decreased respectively by 2.6 and 3.7 percentage points compared to M2 and F2. M4 and F4 were the models used for the two scenarios considering that the drop in performance compared to the previous, more complex models, was minimal.

S11 Table shows the results of our models' predictions at country-level and per sex. For males, Malawi has the lowest predictive power with a F1 score of 61.4% compared to 81.8% for Angola. For females, Angola has the lowest F1 score with 61.8% versus 80.0% for Burundi. Sensitivity values are ranging from 51.3% for males in Malawi to 79.3% for females in Lesotho. For PPV, the lowest value is for males in Burundi with 75.0% versus 100% for males in Angola and Ethiopia.

Again, at country level, we have then aggregated the HIV status predictions per country to estimate national prevalence. Our models underestimated country-specific HIV prevalence but with small relative differences ranging from -0.6% for females in Zimbabwe to -33.3% for males in Malawi (S12 Table). Fig 4 shows two maps per sex, one representing the predicted prevalence per country (left) and the other one the absolute difference between the predicted and the observed prevalence (right). The worst absolute difference is for males in Zambia with -3.1% versus -0.1% for males in Burundi and female in Zimbabwe.

## Scenarios

**1) 95% PLHIV know their status.**   For males, a sensitivity of 95% would require that 5,450 individuals out of 11,031 (49.4%) would need to be tested to identify 840 HIV positives out of the 883 PLHIV. The corresponding PPV is 15.4%; 7 individuals would therefore need to be tested to find one HIV positive person (number needed to test NNT). For females, 6,696 individuals out of 13,926 (48.1%) would need to be tested to find 1,522 HIV positives out of the 1,602 PLHIV. The PPV is 22.7% and the NNT is 5.

**2) At least 90% probability of being HIV positive.**   Out of 11,031 males and 13,926 females, 512 males (4.6%) and 837 females (6.0%) were identified as high-risk populations (i.e. at least 90% of being HIV positive). Overall, 492 males would have been correctly identified as HIV positive out of the 883 male PLHIV (sensitivity of 55.7% and PPV of 96.1%) and 809 females would have been correctly identified as HIV positive out of the 1,602 female PLHIV (sensitivity of 50.5% and PPV of 96.7%).

## Discussion

Using large representative datasets with over 120,000 persons from ten East and Southern African countries, we were able to accurately predict the HIV status of individuals using demographic and socio-behavioural characteristics only. Our approach allowed us to select the nine most important predictor variables common for both sexes: geographic position (longitude, latitude and, altitude), current age, age of most recent partner, total lifetime number of sexual partners, years lived in current place of residence, condom use during last sexual intercourse with most recent partner, and wealth index. Using these nine variables to predict HIV positivity reduces dramatically the amount of knowledge needed to identify key populations.

We also determined the direction of the association between predictor variables and HIV status. We confirmed a number of established HIV risk factors such as older age or older age

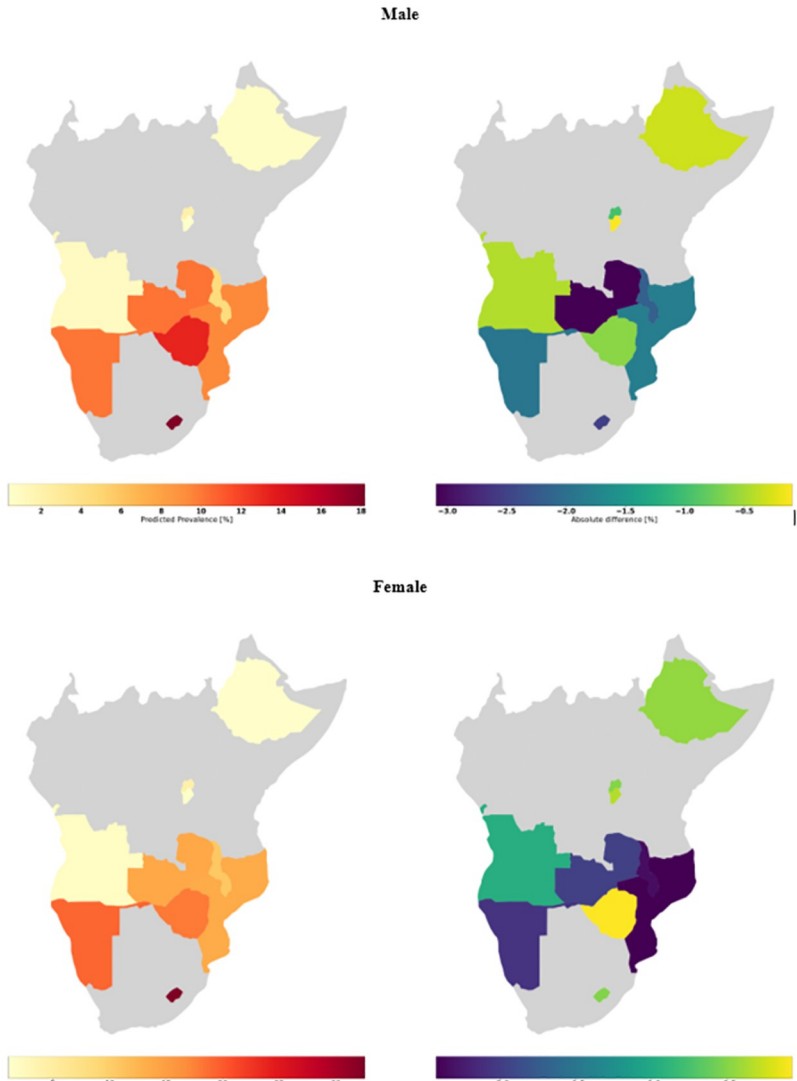

**Fig 4. Predicted prevalence per country (LH) and absolute difference between predicted and observed prevalence (RH).**

of the most recent partner [29], a high number of sexual partners [30], and living in an urban area [31]. Additionally, circumcision and breastfeeding were associated with a lower risk of HIV positivity [31]. Unlike previous findings [32], condom use during the last sexual intercourse increased the probability of HIV positivity in our study. This seemingly counterintuitive finding may be the result of increased condom use in individuals who are already aware of their positive HIV status. The differences in individual HIV status due to the altitude are likely multifactorial. These factors stem from environmental, biological, as well as socio-behavioural and policy-level differences that impact infection and transmission [30, 33–35]. The cross-sectional nature of our study limits our ability to investigate this further. We also identified risk factors for HIV infection which have rarely been investigated before. For example, an increased distance to water source was associated with HIV status; the association could be either positive or negative, but not neutral. A previous study showed that the risk of sexual assault of women, and hence the risk of HIV infection, increased when the time to reach a

water source increased [32]. However, longer time to get to water sources are more common in rural areas where HIV prevalence is known to be generally lower, hence a decrease in risk of HIV positivity.

Our model was also able to accurately predict the prevalence at country level per sex. The difference in predictive power by country depends on many factors, such as the prevalence of the country, the percentage of the country sample size compared to the overall sample and the similarity of the country risk factors versus its peers.

When adapting our predictive algorithm to finding 95% of PLHIV, we needed to test 7 males (NNT of 7; PPV of 15.4%), and 5 females (NNT of 5; PPV of 22.7%) to find one HIV positive individual. A previous systematic review of different testing strategies showed that NNTs ranged between 3 and 86 for community-based testing strategies and between 4 and 154 for facility-based testing strategies [36]. Our method is, consequently, among the best performing testing strategies and can reduce by two the number of tests needed to find 95% of PLHIV compared to current general population testing.

When targeted HIV case-finding strategies are implemented to increase the cost-effectiveness of testing, a high yield is important to ensure that many of those tested are HIV positive. It is currently unknown if additional behavioural-based testing strategies can enhance or complement current targeted case-finding strategies such as index testing. Acceptable cut-offs for both sensitivity and PPV would need to be adapted for specific low resources settings and for the desired testing coverage. In our second scenario, we identified about 5% of the population at high risk of being HIV positive using a probability cut-off of 90%. This allowed us to identify more than 50% of all PLHIV with most of the tested population being HIV-positive; the remaining HIV-negative tested individuals are choice candidates for preventative services such as pre-exposure prophylaxis (PrEP). We were consequently able to maximise the efficacy of the testing. We believe that our method would, therefore, be a valuable addition to current targeted strategies.

To our knowledge, this study is the first to use machine learning methods to predict HIV in generalised HIV epidemic East and Southern African countries using routinely collected survey data. The main scope was to determine common risk factors of HIV positivity between countries with high HIV prevalence and the predictive ability of machine learning models based on these common risk factors. Hence, one of the limitations of this study was the generalizability of our predictive models for countries that were not used to train the algorithm. The accuracy of the prediction decreased, probably due to different risk factor distributions between countries. Future studies could improve the generalizability by selecting more similar countries than the country we aimed at generalizing to or apply our algorithm to country-specific individuals. We were also limited by the available variables in our dataset, and as a result we were unable to consider differences in viral load suppression, health-care expenditure, specific HIV-related interventions, and conflicts and wars. Additionally, missing values were to be found in the data and implied making assumptions about their randomness and using imputation methods that are necessarily imperfect by nature [37]. Finally, although HIV testing was laboratory-based and not self-reported, some results were inconclusive and, thus, discarded. A number of variables were self-reported and therefore subject to social desirability and recall bias.

## Conclusions

Using machine learning algorithms, we identified strong predictors of HIV positivity. Our findings may explain the spatial variability of HIV prevalence and can inform HIV testing strategies in resource-limited settings. While the implementation of a machine learning based

risk score for targeted interventions was feasible in rural East Africa [38], the acceptability and use of potentially sensitive behavioural risk factors to directly identify individuals for HIV testing needs to be evaluated. Our algorithm performed well with only a limited number of variables, which do not require extensive interviews or questionnaires. This approach may be implemented by clinicians and community health care workers or utilised through additional HIV case-finding modalities such as call centres, social media, and self-testing initiatives. The availability of individual-level data on the association of various diseases with socio-behavioural characteristics is rapidly increasing. Advanced methods to analyse these large sources of data can help to prevent, diagnose and treat HIV and other diseases more efficiently.

## Supporting information

**S1 File.**
(DOCX)

**S1 Fig. Values of the F1 score and the Brier score for each of the 50 sets of parameters per algorithm (female ex-Zambia dataset).**
(DOCX)

**S1 Table. List of the Demographic and Health Surveys (DHS) survey year.**
(DOCX)

**S2 Table. Data processing.**
(DOCX)

**S3 Table. Summary statistics of the observed and imputed data for the incomplete variables in the male test dataset.**
(DOCX)

**S4 Table. Summary statistics of the observed and imputed data for the incomplete variables in the female test dataset.**
(DOCX)

**S5 Table. List of variables.**
(DOCX)

**S6 Table. Characteristics of Demographic and Health Survey (DHS) individuals.**
(DOCX)

**S7 Table. Results of the XGBoost algorithm per sex for the validation (80%; 5-fold cross-validation), test (20%) and, left-out (excluded country) samples.**
(DOCX)

**S8 Table. Results of the Support Vector Machine (SVM) algorithm per sex for the validation, test and, left-out samples.**
(DOCX)

**S9 Table. Results of the ElasticNet algorithm per sex for the validation, test and, left-out samples.**
(DOCX)

**S10 Table. Results of the Generalized Additive Model (GAM) algorithm per sex for the validation, test and, left-out samples.**
(DOCX)

**S11 Table. F1, sensitivity, PPV and Brier score per country for models M4 and F4.**
(DOCX)

**S12 Table. Predicted prevalence per country.**
(DOCX)

## Acknowledgments

We thank Antoine Flahault, Danny Sheath, and Isotta Triulzi for helpful discussions and comments.

## Author Contributions

**Conceptualization:** Olivia Keiser.

**Formal analysis:** Erol Orel.

**Investigation:** Erol Orel.

**Methodology:** Erol Orel, Aziza Merzouki.

**Supervision:** Stéphane Marchand-Maillet, Aziza Merzouki, Olivia Keiser.

**Validation:** Erol Orel, Stéphane Marchand-Maillet.

**Writing – original draft:** Erol Orel.

**Writing – review & editing:** Erol Orel, Rachel Esra, Janne Estill, Amaury Thiabaud, Stéphane Marchand-Maillet, Aziza Merzouki, Olivia Keiser.

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
