## [Decision Letter · Decision Letter 0]

30 Jul 2021

PONE-D-21-10249

Prediction of HIV status based on socio-behavioural characteristics in East and Southern Africa

PLOS ONE

Dear Dr. Orel,

Thank you for submitting your manuscript to PLOS ONE. After careful consideration, we feel that it has merit but does not fully meet PLOS ONE’s publication criteria as it currently stands. Therefore, we invite you to submit a revised version of the manuscript that addresses the points raised during the review process.

We look forward to receiving your revised manuscript.

Kind regards,

Daniel Boateng

Academic Editor

PLOS ONE

Journal Requirements:

“We acknowledge the support of the Swiss National Science Foundation (SNF professorship grant n° 163878 to O Keiser) which funded this study. “

 “We acknowledge the support of the Swiss National Science Foundation (SNF professorship grant n° 163878 to O Keiser) which funded this study. he funders had no role in study design, data collection and analysis, decision to publish, or preparation of the manuscript.”

Additional Editor Comments (if provided):

The article addresses a very important subject in Global health. However, both reviewers raised a important methodological concerns that must be addressed and resubmitted for consideration.

Reviewers' comments:

Reviewer's Responses to Questions

**Comments to the Author**

1. Is the manuscript technically sound, and do the data support the conclusions?

Reviewer #1: Partly

Reviewer #2: Yes

2. Has the statistical analysis been performed appropriately and rigorously? 

Reviewer #1: I Don't Know

Reviewer #2: Yes

3. Have the authors made all data underlying the findings in their manuscript fully available?

Reviewer #1: No

Reviewer #2: No

4. Is the manuscript presented in an intelligible fashion and written in standard English?

Reviewer #1: Yes

Reviewer #2: Yes

5. Review Comments to the Author

Reviewer #1: This is an interesting and timely application of machine learning to a problem of substantial societal impact. Unfortunately there are shortcomings in the description of the methods and potentially also methodological issues which I fear invalidate the results (for these reasons cannot judge whether statistical analyses are performed appropriately). Please see attached document for my detailed evaluation of this paper.

Reviewer #2: Reviewer comments on “Prediction of HIV status based on socio-behavioural characteristics in East and Southern Africa”

Overall, this is an interesting paper that explores different machine learning methods (algorithms) to predict individual-level HIV status in East and Southern Africa. Although the work could be relevant to the global public health field, more emphasis should be placed on (i) exploration of model accuracy/predictive power by country, African region and at smaller scales within countries, (ii) comparison with previous studies that looked at underlying predictors of HIV prevalence, and (iii) judging the suitability of such models for areas where no or scarce HIV data are available and what this depends on.

The authors can further elaborate on the difference in prediction accuracy by country and by African region: now, the same variables are used for the predictions, but mechanisms underlying HIV trends and risks might be highly country-specific. Does modelling by country lead to the same selection of the 10 most predictive variables? And how much does the predictive power differ by country, and what would this depend on? Also, would it be possible to use the algorithms with the same 10 variables to make predictions for smaller (subnational) areas? So that policy makers could estimate the HIV prevalence in their area when outdated or unreliable HIV data are available.

Would it be possible to add maps of the measured versus predicted HIV prevalence by country or province, and/or a map with the discrepancy in measured versus predicted prevalence? The current figures are nice, but quite technical, it would be good to add one or two plots or maps that are easier to interpret without a lot of technical knowledge, the enhance usefulness to policy makers.

Please further explain in the discussion why the 10 most important variables could logically be predictive for HIV status and compare your findings with those of previous studies that looked at predictive variables for HIV status in sub-Saharan Africa, such as:

Palk, Laurence, and Sally Blower. "Geographic variation in sexual behavior can explain geospatial heterogeneity in the severity of the HIV epidemic in Malawi." BMC medicine 16.1 (2018): 1-9.

Dwyer-Lindgren, Laura, et al. "Mapping HIV prevalence in sub-Saharan Africa between 2000 and 2017." Nature 570.7760 (2019): 189-193.

Bulstra, Caroline A., et al. "Mapping and characterising areas with high levels of HIV transmission in sub-Saharan Africa: A geospatial analysis of national survey data." PLoS medicine 17.3 (2020): e1003042.

For some predictive variables in particular, the predictive mechanisms are not obvious. For example, how does altitude affect HIV status, and are latitude, longitude and altitude included separately? The charm of the applied machine learning algorithms is that you let the data speak, but it would be informative if the authors think about potential mechanisms in which the variables affect HIV status. They might be proxy variables for other, more intuitive, variables of impact.

Some further explanation is needed on splitting the data between testing and training dataset; was this done at the individual level or at the sample location level? And was the 80%-20% selection present for every country, or only for the overall data sample?

In the abstract, I would suggest to add the included countries to the abstract, instead of only “10 countries in East and Southern Africa” and explain what the F1 score is, since reader might not be familiar with this.

6. PLOS authors have the option to publish the peer review history of their article (what does this mean?). If published, this will include your full peer review and any attached files.

Reviewer #1: No

Reviewer #2: **Yes: **Caroline A. Bulstra

---

## [Author Response · Author response to Decision Letter 0]

8 Nov 2021

Dear editor, 

Thank you very much for the helpful reviews which allowed us to substantially improve the manuscript. Please find below our answers to the referee comments.

Yours sincerely,

Erol Orel, Aziza Merzouki and Olivia Keiser on behalf of all co-authors.

Comments

Editorial Comments to Authors

Journal Requirements:

The manuscript meets PLOS ONE’s requirements, including those for file naming.

“We acknowledge the support of the Swiss National Science Foundation (SNF professorship grant n° 163878 to O Keiser) which funded this study. “

Please remove any funding-related text from the manuscript and let us know how you would like to update your Funding Statement.

Any funding-related text has been removed from the Acknowledgements section.

Currently, your Funding Statement reads as follows: “We acknowledge the support of the Swiss National Science Foundation (SNF professorship grant n° 163878 to O Keiser) which funded this study. he funders had no role in study design, data collection and analysis, decision to publish, or preparation of the manuscript.” Please include your amended statements within your cover letter; we will change the online submission form on your behalf.

Amended Funding Statement: “We acknowledge the support of the Swiss National Science Foundation (SNF professorship grant n° 163878 and grant n° 202660 to O Keiser) which funded this study. The funders had no role in study design, data collection and analysis, decision to publish, or preparation of the manuscript.”

Additional Editor Comments (if provided):

The article addresses a very important subject in Global health. However, both reviewers raised important methodological concerns that must be addressed and resubmitted for consideration.

Reviewer 1

In this manuscript, the authors use publicly available socio-behavioural data from the Demographic and Health Surveys program for 10 countries in Eastern and Southern Africa to predict HIV status for individuals using machine learning and statistical learning models. The authors first compared four different types of model types (penalized logistic regression, generalized additive model, support vector machines and gradient-boosted trees) to identify the best-performing type (gradient-boosted trees) according to the so-called F1 metric, which is a well-known but less transparent metric. Subsequently the authors then identified the subset of most contributing features and built new models of the best type around these. Based on these models, the authors then investigated two scenarios for targeting preventive action: a) the number of persons to be tested to reach a sensitivity of 95 % (true positive rate), and b) identified a population for which the probability of being HIV positive was higher than 95 %.

This is an area of significant importance, and indeed an area in which modern machine learning methods can be brought to bear for great societal impact. The paper is furthermore well-written and mostly easy to read, and the figures and tables are generally well-prepared (more on this below). It therefore saddens me to report that I cannot support publication at this stage due to serious shortcomings of the methods description, and potentially also inconsistencies in the results which undermine the claims of the authors (this latter I cannot judge in full due to shortcomings of the methods description).

Major points

1. The methods section is missing a number of important aspects in order for me to assess in detail the steps the authors have taken. How were the models fitted?

Each algorithm was assessed using a stratified 5-fold cross-validation on 50 sets of hyperparameters values, same metrics measured (namely F1, sensitivity, positive predictive value (PPV) and Brier score), same threshold used (0.5), and best model selection based on highest F1 on the test dataset. What differs is the algorithms and their respective objective functions (training loss + regularization) together with the number and type of the hyperparameters. We have added the following information as a section named “Models” in the Supplementary Material:

Generalized additive models (Logistic GAM)

GAM takes the functional form above with in our case the intercept set to zero. Since the scale of the Binomial distribution is known, our gridsearch minimizes an Un-Biased Risk Estimator (UBRE) objective: where D is the deviance, n the number of data, s the scale parameter (equal to 1 in our case) and DoF the effective degrees of freedom of the model. The feature functions are penalized B splines (P spline) with 20 basis functions for each by default. The smoothing hyperparameter lambda is drawn from a continuous uniform distribution between exp(-3) and exp(3) for each of the spline functions.

Penalized Logistic Regression (Elastic Net)

Logistic regression takes the above functional form with in our case the Intercept set to zero. The two hyperparameters of the loss function below were evenly distributed on a logarithmic space (C: logspace(-9, 9), l1ratio: logspace(-9, 0)):

The parameter C is the inverse of the regularization strength alpha and l1ratio is the Elastic-Net mixing regularization hyperparameter where l1ratio=0 is equivalent to using a l2 penalty and l1ratio=1 equivalent to using a l1 penalty .

Support Vector Classifier (SVC)

The radial basis function kernel has been used for this learning algorithm. The parameter C (evenly distributed on a logarithmic space (logspace(-9, 9)) is the regularization hyperparameter, similar to the one of Elastic Net (i.e. the strength of the regularization is inversely proportional to C) of the below loss function. The penalty is a squared l2 penalty:

Extreme Gradient Boosting (XGBoost)

Nine hyperparameters were included into the random grid search. The details of each parameter space are:

The objective function is: 

where in our case, l is the log likelihood of the Bernoulli distribution (i.e. log loss). For a more detailed overview of the regularization term and all the hyperparameters you can refer to the following article by Chen & Guestrin: https://arxiv.org/pdf/1603.02754.pdf.

What were the target variables, and what was the outcome from each of the trainings?

The target variable was the HIV status of the individuals (0 for HIV negative and 1 for HIV positive). We have now added this explicitly line 15 of the Methods section.

The outcome from each of the training for each of the algorithms can be found on the Gitlab repository https://gitlab.com/Triphon/predicting_hiv_status, in the ““scripts/algorithm_name/” folder. It consists of the precision, recall, F1 and Brier score obtained for the 50 sets of hyperparameters on each 5 validation samples, for each of the 20 datasets (2 sex and 10 left-one-country datasets) stored in a joblib format.

As far as I can tell from the files shared in the author’s model and code repository the model selection analysis is not included (and the readme in the repository is not providing much help).

Each algorithm’ results can be found tables A6i to A6iv in the “Supplementary Material” and can be extracted from each “scripts/algorithm_name” folders using the algorithm_name.ipynb jupyter notebook. The selection between the 4 algorithms was then simply done by averaging the F1 scores of the 10 test datasets by sex and ranking the obtained values by descending order.

Regarding the variables selection in the second part of the analysis, the scripts and the obtained results of the SFFS procedure can be found in the “scripts/variable_selection” folder (“sffs.ipynb”). The final model with only 9 variables is available in the “scripts/xgb” folder and named xgb_9.ipynb.

We have simplified the structure of the repository and rewritten the readme file to better explain this.

As will be apparent further below, details of model training will be important to assess the validity of the results the authors claim. Description of the details of the imputation methods used are missing. Is the variance conserved for the MICE algorithm that the authors use?

We expanded the description of the MICE imputation method which can be found in the Supplementary Material at the “MICE imputation” section and referenced in the main manuscript in the Methods at line 19. An in-depth overview on how the imputation was conducted can be found in the script “scripts/data_processing_engineering/imputation.py” in the Gitlab repository. Two descriptive statistics tables (mean, standard deviation, minimum and maximum) of observed and imputed data for male and female test datasets have been added to the Supplementary Material (Table 5i and Table 5ii). For a detailed description of the XGBoost “imputation” method (indeed more a method for dealing with missing values), please refer section 3.4. of the XGBoost article https://arxiv.org/pdf/1603.02754v2.pdf. This reference was cited in the main manuscript (reference 23) at line 62 of the Methods section.

And what are the implications of using the imputation of the XGBoost algorithm? This needs to be clarified to know whether this step is artificially limiting or enhancing the presented results.

The two methods were giving very similar results and the implications of using the XGBoost algorithm with missing values were mainly practical. First, the computational time of MICE when dealing with a high number of variables (84 and 122 respectively for male and female) was approximately 3 days using the High-Performance Computing resources (Baobab cluster) of the University of Geneva against few hours when using the XGBoost algorithm. Also, hundreds of lines of code (see “scripts/data_processing_engineering/imputation.py”) were needed to perform the MICE imputation where no additional coding was needed with the XGBoost algorithm. We have rephrased it in the main manuscript in the Results section at line 33.

I cannot follow the details of the three-step training process (Fig. 1). Is the whole thing a nested cross-validation where the outer loop (Step 1) is across countries (switching “holdout country”) and the inner step (Steps 2 and 3) is the 5-fold cross-validation used during model training? I sense the approach is fine but cannot follow all steps to make sure. 

From our entire datasets for males and females, we first left one of the 10 countries out (switching left out country) to create 10 different datasets per sex comprised of only 9 countries. This has been done for generalization purposes in order to assess the quality of our models when the data were not drawn from the exact same distribution. Then, each of the 10 newly created datasets per sex were split between a stratified (due to imbalanced outcomes) 80% training set and a 20% test set (Fig. 1 Step. 1). A stratified 5-fold cross-validation was then performed for each algorithm on each of the training sets for training and validation (Step. 2). This could effectively be seen as the inner step in a classical nested cross-validation. Each of the 10 best models per sex and per algorithm was then tested on the corresponding test set and the resulting F1 scores were averaged (first part of Step. 3) similar to the outer step. Finally, we applied each selected model on the corresponding left out country dataset (second part of Step. 3). We have rewritten the Methods section for better clarity (see line 23 to 34).

Figures and tables describing data preparation and model training are largely missing captions. This makes it harder to follow the steps; in cases like Table A2 I simply cannot infer how to read the table. What does each row represent? What happens between each row?

The different steps of the data preparation can be found in the Supplementary Material (see line 1 to 24). We have modified Table A2 in the Supplementary Material to make it self-explanatory and more readable.

2. Two of the four model types, specifically support vector machines and gradient-boosted trees, actually do not model the probability that an individual has HIV unless specific steps are taken (e.g. use Platt scaling). It does not seem to be the case that such steps have been taken but cannot know for sure due to insufficient methods details provided. It is highly problematic that gradient-boosted trees (the type which is selected by the authors) does not return probabilities since the authors use it to identify the subpopulation which has more than 95 % probability of having HIV (scenario 2); put differently, the results from scenario 2 cannot be trusted since gradient-boosted trees are used. This concern seems to invalidate these results. For context: These models assign individuals to classes based on classification rules (the sign of the solution to the convex quadratic optimization problem for support vector machines, the particular splitting rules used in the gradient-boosted trees (usually cross-entropy) [1]) instead of modeling directly the probability that an individual has HIV. These are examples of improper scoring rules which are well-known to cause incorrect probabilities. An observed probability can be inferred afterwards from these models, but that will depend on the particular elements of the dataset and as such will change with the addition of a single new datapoint, and is in general not a good estimate of the actual probability.

We have now calibrated the predicted probabilities obtained by XGBoost using a sigmoid (Platt scaling) and incorporated these changes into the Results section line 87 to 93. The differences in scenario 2 resulting from the calibration are as follows: 1) The probability threshold has been moved from 95% to 90%. 2) Without calibration, out of 11,031 males and 13,926 females, 551 males (5.0%) (461 previously i.e. 4.2%) and 1’113 females (8.0%) (862 previously i.e. 6.2%) were identified as high-risk populations. Overall, 526 males (447 previously) would have been correctly identified as HIV positive out of the 883 male PLHIV (sensitivity of 59.6% (50.6% previously) and PPV of 95.5% (97.0% previously)) and 1’065 females (833 previously) would have been correctly identified as HIV positive out of the 1,602 female PLHIV (sensitivity of 66.5% (52.0% previously) and PPV of 95.7% (96.6% previously)). 3) With calibration, 512 males (4.6%) and 837 females (6.0%) were identified as high-risk populations. Overall, 492 males would have been correctly identified as HIV positive (sensitivity of 55.7% and PPV of 96.1% ) and 809 females would have been correctly identified as HIV positive (sensitivity of 50.5% and PPV of 96.7%).

3. The results in Fig. 2 illustrate substantially lower F1 scores on the left-out samples for support vector machines and gradient-boosted trees (XGBoost) which need to be investigated. This illustrates that the performance found in training is not at all retrieved when the models are applied to new data, i.e. it undermines the trust which can be put in the abilities of these models to offer usable predictions. This puts all presented results at risk of being incorrect! While the authors do observe this, no explanation is offered nor is further investigation conducted. As a minimum the authors would need to explain why this should not be a point of concern for trusting the results. It is unclear to me whether this is a result of overfitting, is a consequence of using the F1 metric (itself an improper scoring rule) to evaluate the models (that metric itself being a nonlinear function of the class assignments of the models), or something else entirely.

We believe this should not be a point of concern for trusting the results because, unlike the test sets (for which the results are similar to the cross-validation), the left-out countries are not drawn from the same probability density function as the training sets. The purpose of this was to study the generalization of our model on data distributions that are different from the one used to train and validate the model. In the Discussion section, the following explanation was given line 51: “Hence, one of the limitations of this study was the generalizability of our predictive models for countries that were not used to train the algorithm. The accuracy of the prediction decreased, probably due to different risk factor distributions between countries. Future studies could improve the generalizability of our models by training them on more similar countries than the country we aimed at generalizing to.

4. It appears the authors are using the F1 score to pick the best model (in Algorithms, as part of Methods section). As already mentioned, this is an improper scoring rule (which does not indicate which model best predicts the probability that an individual has HIV) and selecting models based on this metric could lead to models with good F1 scores which are a bad representation of whether or not an individual has HIV! The authors will need to clarify that they are not selecting models based on an improper scoring rule. More details on considerations for the important topic of proper scoring rules for classification can be found in [2] and [3].

While we understand the reviewer’s point of concern about F1 score being an improper scoring metric, not based on probability and depending on an arbitrary threshold (“business” decision), we decided to use this metric for three main purposes: 1) ability to compare our results with previous studies; 2) appropriate metric for imbalanced datasets; 3) importance of targeting specific sensitivity and PPV values to achieve different testing strategies. This is explained in the sub-section Algorithms of the Methods section from line 8 to line 13. However, in order to compare our results with proper scoring rules, we have recomputed entirely the part of our analysis comparing the 4 algorithms using the Brier score and populated Tables A6i to A6iv in the Supplementary Material section with its values. XGBoost is still the best performing algorithm, followed by SVM, GAM and finally ElasticNet. In general, the scores obtained by the models with the best Brier score were very similar to the ones obtained with the best F1 score. We have added some sentences about Brier scores results in the Results section from line 14 to line 17.

Minor points

1. The authors use random selection of model hyper-parameters, which is a well-established approach. It would be good with some graphical illustration of the improvement from hyperparameter optimization as an insight into whether model type or hyperparameter tuning is more important for this case.

We have added in the Supplementary Material a graphical illustration (Figure A1) showing the F1 scores per algorithm obtained with the 50 sets of parameters for one of the female datasets with 9 countries (ex-Zambia) as a case example. In addition, we have also compared it with the Brier scores obtained.

2. The authors use Shapley values to assess the impact of each covariate on the outcome, which is also an established approach. However, readers without deep statistical training could be led to believe that the type of impact suggested by Shapley values could be used for shaping intervention strategies. Unfortunately, this would not be correct because Shapley values do not describe the causal impact of each covariate, only the additional change in overall outcome by adding this covariate. I would suggest adding a comment along these lines to mitigate any unintended conclusions.

A comment has been added at the top of the Shapley values graph to explain these limitations. “It has to be noted here that Shapley values do not describe the causal impact of each covariate, only the additional change in overall outcome by adding this covariate.”

Reviewer 2

Overall, this is an interesting paper that explores different machine learning methods (algorithms) to predict individual-level HIV status in East and Southern Africa. Although the work could be relevant to the global public health field, more emphasis should be placed on (i) exploration of model accuracy/predictive power by country, African region and at smaller scales within countries, (ii) comparison with previous studies that looked at underlying predictors of HIV prevalence, and (iii) judging the suitability of such models for areas where no or scarce HIV data are available and what this depends on.

The authors can further elaborate on the difference in prediction accuracy by country and by African region: now, the same variables are used for the predictions, but mechanisms underlying HIV trends and risks might be highly country-specific. Does modelling by country lead to the same selection of the 10 most predictive variables? 

The main scope of this study was to determine common risk factors of HIV positivity between countries with high HIV prevalence and the predictive ability of machine learning models based on these common risk factors. Hence, we did not train our algorithm with country-specific individuals. However, although this was not in the scope of this paper, we now mention it in the Discussion section line 55 as something worth exploring in the future.

And how much does the predictive power differ by country, and what would this depend on?

We computed the F1 score, sensitivity and PPV values separately for each of the 10 countries, as obtained by the model. We have created Table A7 and added it in the Supplementary Material. The difference in predictive power by country would depend on many factors, some of them being: the prevalence of the country, the percentage of the country sample size compared to the overall sample, the similarity of the risk factors between the country and its peers, etc...

Would it be possible to add maps of the measured versus predicted HIV prevalence by country or province, and/or a map with the discrepancy in measured versus predicted prevalence? 

Yes, the prediction of individuals HIV status can be pooled by country in order to compute predicted prevalence. We have added Table A8 in the Supplementary Material which shows the predicted versus the measured prevalence per country, the absolute and the relative differences between both. In addition, we have produced 2 maps per sex showing the predicted prevalence per country and the absolute differences. We have incorporated these maps in the main document by replacing the previous Figure 4. We have reported the results in the Results section from line 73 and discussed these findings in the Discussion section from line 28.

Also, would it be possible to use the algorithms with the same 10 variables to make predictions for smaller (subnational) areas? So that policy makers could estimate the HIV prevalence in their area when outdated or unreliable HIV data are available.

Yes, by the same methodology used to compute predicted prevalence at country level, we could compute predicted prevalence at district level. 

The current figures are nice, but quite technical. It would be good to add one or two plots or maps that are easier to interpret without a lot of technical knowledge, to enhance usefulness to policy makers.

Following your advice, we have replaced the quite technical Figure 4 with the maps discussed above. We have also removed Figure 3A and 3B for the sake of simplicity and better clarity. We were unfortunately limited to five figures and tables in total, hence, we have added in the Supplementary Material section Table A7 showing the different values of the F1 score, sensitivity and PPV per country and Table A8 highlighting the difference between predicted and measured prevalence per country.

Please further explain in the discussion why the 10 most important variables could logically be predictive of HIV status and compare your findings with those of previous studies that looked at predictive variables for HIV status in sub-Saharan Africa, such as:

- Palk, Laurence, and Sally Blower. "Geographic variation in sexual behavior can explain geospatial heterogeneity in the severity of the HIV epidemic in Malawi." BMC medicine 16.1 (2018): 1-9.

- Dwyer-Lindgren, Laura, et al. "Mapping HIV prevalence in sub-Saharan Africa between 2000 and 2017." Nature 570.7760 (2019): 189-193.

- Bulstra, Caroline A., et al. "Mapping and characterising areas with high levels of HIV transmission in sub-Saharan Africa: A geospatial analysis of national survey data." PLoS medicine 17.3 (2020): e1003042.

We have further explained in the Discussion section (from line 10 to line 27) how the most important variables found by our method could logically be predictive of individual HIV status and compared it with previous studies on risk factors. We have included in these comparisons the studies you have kindly brought to our attention and these papers have been added to the References section. 

The charm of the applied machine learning algorithms is that you let the data speak, but it would be informative if the authors think about potential mechanisms in which the variables affect HIV status. They might be proxy variables for other, more intuitive, variables of impact. For some predictive variables in particular, the predictive mechanisms are not obvious. For example, how does altitude affect HIV status, and are latitude, longitude and altitude included separately?

We have added the following sentence line 17 in the Discussion section: “The differences in individual HIV status due to the altitude are likely multifactorial. These factors stem from environmental, biological, as well as social and policy-level differences that impact infection and transmission.” Yes, altitude, latitude and longitude have been included separately, however, algorithms based on decision trees are able to discover interaction among independent variables.

Some further explanation is needed on splitting the data between testing and training dataset; was this done at the individual level or at the sample location level?

The split of data between training and testing has been done at the individual level. We have added this precision in the Method section line 26.

And was the 80%-20% selection present for every country, or only for the overall data sample?

The 80/20 split has been performed on every 10 subsets per sex comprised of 9 countries in the first part of our analysis and on the overall data samples per sex in the second part of our analysis.

In the abstract, I would suggest adding the included countries, instead of only “10 countries in East and Southern Africa” and explain what the F1 score is, since readers might not be familiar with this.

We have added countries' names in the Introduction section line 3 of the Abstract and a definition of the F1 score in the Methods section line 5 of the Abstract.

---

## [Decision Letter · Decision Letter 1]

15 Dec 2021

PONE-D-21-10249R1Prediction of HIV status based on socio-behavioural characteristics in East and Southern AfricaPLOS ONE

Dear Dr. OREL,

Thank you for submitting your manuscript to PLOS ONE. After careful consideration, we feel that it has merit but does not fully meet PLOS ONE’s publication criteria as it currently stands. Therefore, we invite you to submit a revised version of the manuscript that addresses the points raised during the review process. Please submit your revised manuscript by Jan 29 2022 11:59PM. If you will need more time than this to complete your revisions, please reply to this message or contact the journal office at plosone@plos.org. Please include the following items when submitting your revised manuscript:A rebuttal letter that responds to each point raised by the academic editor and reviewer(s). You should upload this letter as a separate file labeled 'Response to Reviewers'.A marked-up copy of your manuscript that highlights changes made to the original version. You should upload this as a separate file labeled 'Revised Manuscript with Track Changes'.An unmarked version of your revised paper without tracked changes. You should upload this as a separate file labeled 'Manuscript'.If applicable, we recommend that you deposit your laboratory protocols in protocols.io to enhance the reproducibility of your results. Protocols.io assigns your protocol its own identifier (DOI) so that it can be cited independently in the future. For instructions see: https://journals.plos.org/plosone/s/submission-guidelines#loc-laboratory-protocols. Additionally, PLOS ONE offers an option for publishing peer-reviewed Lab Protocol articles, which describe protocols hosted on protocols.io. Read more information on sharing protocols at https://plos.org/protocols?utm_medium=editorial-email&utm_source=authorletters&utm_campaign=protocols.

We look forward to receiving your revised manuscript.

Kind regards,

Daniel Boateng

Academic Editor

PLOS ONE

Journal Requirements:

Reviewers' comments:

Reviewer's Responses to Questions

**Comments to the Author**

1. If the authors have adequately addressed your comments raised in a previous round of review and you feel that this manuscript is now acceptable for publication, you may indicate that here to bypass the “Comments to the Author” section, enter your conflict of interest statement in the “Confidential to Editor” section, and submit your "Accept" recommendation.

Reviewer #1: (No Response)

Reviewer #2: All comments have been addressed

2. Is the manuscript technically sound, and do the data support the conclusions?

Reviewer #1: Partly

Reviewer #2: Yes

3. Has the statistical analysis been performed appropriately and rigorously? 

Reviewer #1: No

Reviewer #2: Yes

4. Have the authors made all data underlying the findings in their manuscript fully available?

Reviewer #1: Yes

Reviewer #2: Yes

5. Is the manuscript presented in an intelligible fashion and written in standard English?

Reviewer #1: Yes

Reviewer #2: Yes

6. Review Comments to the Author

Reviewer #1: The authors have addressed nearly all the comments I raised in the first review, and have done so in satisfactory fashion. There seems to be one open point on the statistical methods related to their imputation methods, which I think should be addressed before the paper can be accepted for publication. I also feel the More details in attached document.

Reviewer #2: The authors have rigorously approved the methods descriptions in the manuscript and have added several sensitivity analyses and visualisations to further improve understanding of the results for the general public. My recommendation is to accept the paper for publication in PLOS One.

7. PLOS authors have the option to publish the peer review history of their article (what does this mean?). If published, this will include your full peer review and any attached files.

Reviewer #1: No

Reviewer #2: No

---

## [Author Response · Author response to Decision Letter 1]

31 Jan 2022

Reviewer 1

I am happy to see that the reviewers have addressed most of my comments. In their reply, they provide satisfactory replies and they have made good changes to the manuscript, supplement and code repository which makes it easier to follow. I am also happy to see the new Fig 4 which is a great addition.

Unfortunately, there are still two outstanding points which I think should be further elaborated.

1) The authors have added description of the MICE imputation as I requested, but it is unfortunately not clear from this description if steps have been taken to ensure that the imputation does not affect the variance of the imputed variables. If no steps are taken, imputation can influence the distribution of the imputed variable (often reducing the variance, sometimes also affecting the mean), which will likely impact the results (e.g. influence the ability to provide accurate predictions). Judging from the first 10-15 rows in Table S3i it seems that the imputed variables have lower variance. The authors should clarify which steps have been taken to ensure that variance and mean have not been affected by the imputations.

Many thanks, we are happy that the important work done to address the comments and revise the pa-per is acknowledged and valued. Regarding MICE imputation, imputing missing data has always been a challenge, and as stated in the reviewer’s comment, results of any statistical analysis can be only as good as the quality of the data (garbage in - garbage out) and this is particularly true in presence of missing data. In our specific case, we have assumed that the missing values were Missing At Random (MAR) and used the MICE imputation method (similar to the flowchart below from Jakobsen, J.C., Gluud, C., Wetterslev, J. et al. When and how should multiple imputation be used for handling miss-ing data in randomised clinical trials – a practical guide with flowcharts. BMC Med Res Methodol 17, 162 (2017). https://doi.org/10.1186/s12874-017-0442-1). We now explicitly mention this assumption in the revised manuscript (line 129) and cite the additional reference paper from Jakobsen et al.

In other words, we assumed that there might be a relationship between the propensity of missing values and the observed data. MAR data is more common than Missing Completely At Random (MCAR) in all disciplines. Where in the case of MCAR, missing and observed observations are generated from the same distribution, in the case of MAR, the missing and observed observations are no longer coming from the same distribution. Nguyen et al. [Nguyen, C.D., Carlin, J.B. & Lee, K.J. Model checking in multiple imputation: an overview and case study. Emerg Themes Epidemiol 14, 8 (2017). https://doi.org/10.1186/s12982-017-0062-6] suggests that these discrepancies between observed and imputed data are not necessarily problematic, since under MAR we may expect such differences to arise.

In order to check the obtained models in MICE and to ensure the validity of our findings, we have taken a multifaceted approach:

- We assessed the plausibility of the imputed data using subject matter knowledge, harmonizing imputed data to the observed data types (e.g. dichotomic) and to the domain of definition or scale (e.g. strictly positive or maximum value). For a complete overview, the python script can be found in the project repository at “scripts/data_processing_engineering/imputation.py”. This reference to the code was added to the Supplementary Material in the MICE imputation section.

- We compared the imputed data with the observed data to identify major problems with the imputation model. As a rule of thumb, Stuart et al. [Stuart EA, Azur M, Frangakis C, Leaf P. Multiple Imputation with large data sets: a case study of the children’s mental health initiative. Am J Epidemiol. 2009;169(9):1133–9.] proposed comparing the means and variances of observed and imputed values and suggested flagging variables if the ratio of variances of the observed and imputed values was less than 0.5 or greater than 2, or if the absolute difference in means was greater than two standard deviations.

- We compared the scores and results obtained with the MICE imputation method to the ones obtained using the XGBoost built-in method and found very similar values. This is presented in main manuscript line 241-248.

- We used MICE imputation in the first part of our analysis, when we compared the predictive performance of the different algorithms. The performance of all algorithms was compared over the same potentially imperfect imputed datasets. We did not use the MICE imputed datasets in the second part of our analysis, i.e. for variables selection and HIV status prediction (used XGBoost built-in method). 

Finally, we have added as a limitation, the following sentence line 376-378 of the “Discussion” section: “Additionally, missing values were to be found in the data and implied making assumptions about their randomness and using imputation methods that are necessarily imperfect by nature.”

2) The authors take several steps during their model identification and training procedure. This methodological rigor is a strong point of the paper. Unfortunately, it can still be hard to follow all the steps the authors take, so I think it could improve the paper even more if the methods section started with an overview of the different steps.

For better clarity, we have rewritten the different steps for model identification and training procedure 

in the “Methods” section by adding a specific sub-section named “Training, validation and test procedure steps” starting line 23.

Furthermore, upon identifying the best model, the authors seem to follow a different process when fitting the selected model in order to draw the conclusions they present (using data from all 10 countries, fitting using a regular five-fold cross validation). I cannot seem to find a description of whether any special considerations have been made to the test sets for this model (which is used to generate the results in the paper) – are they randomly selected? Stratified? It would be good if the authors could comment on this as well.

Thank you for pointing this out. We used the exact same training, validation and testing strategy than in the first part of our analysis except that no country was left out. We have made it more explicit in the “Methods” section starting line 178-181 (beginning of the sub-section “Variables selection and HIV status prediction”):

“For variables selection and HIV status prediction, we used the exact same training, validation and testing strategy than in the first part of our analysis except that no country was left out. We split each unique dataset par sex into a stratified 80% training and validation set and a 20% test set. The best algorithm was trained and validated using a random grid-search over 250 sets of hyperparameters and a stratified 5-fold cross-validation”.

---

## [Editor Report · Decision Letter 2]

11 Feb 2022

Prediction of HIV status based on socio-behavioural characteristics in East and Southern Africa

PONE-D-21-10249R2

Dear Dr. Orel,

We’re pleased to inform you that your manuscript has been judged scientifically suitable for publication and will be formally accepted for publication once it meets all outstanding technical requirements.

Kind regards,

Daniel Boateng

Guest Editor

PLOS ONE
---

## [Editor Report · Acceptance letter]

18 Feb 2022

PONE-D-21-10249R2 

Prediction of HIV status based on socio-behavioural characteristics in East and Southern Africa 

Dear Dr. OREL:

I'm pleased to inform you that your manuscript has been deemed suitable for publication in PLOS ONE. Congratulations! Your manuscript is now with our production department. 

Kind regards, 

on behalf of

Dr. Daniel Boateng 

Guest Editor

PLOS ONE